# Reduction of Influenza A Virus Prevalence in Pigs at Weaning After Using Custom-Made Influenza Vaccines in the Breeding Herds of an Integrated Swine Farm System

**DOI:** 10.3390/v17020240

**Published:** 2025-02-10

**Authors:** Jorge Garrido-Mantilla, Juan Sanhueza, Julio Alvarez, Jeremy S. Pittman, Peter Davies, Montserrat Torremorell, Marie R. Culhane

**Affiliations:** 1Veterinary Population Medicine Department, College of Veterinary Medicine, University of Minnesota, St. Paul, MN 55108, USA; jorge.garrido@altosano.com (J.G.-M.); davie001@umn.edu (P.D.); torr0033@umn.edu (M.T.); 2Departamento de Ciencias Veterinarias y Salud Pública, Facultad de Recursos Naturales, Universidad Católica de Temuco, Temuco 4810399, Chile; jsanhueza@uct.cl; 3Centro de Vigilancia Sanitaria Veterinaria (VISAVET), Universidad Complutense, 28040 Madrid, Spain; 4Departamento de Sanidad Animal, Facultad de Veterinaria, Universidad Complutense, 28040 Madrid, Spain; 5Smithfield Hog Production, Warsaw, NC 28398, USA

**Keywords:** influenza, custom-made vaccines, pigs, surveillance, swine, farms, vaccination

## Abstract

Vaccination is a common influenza A virus (IAV) control strategy for pigs. Vaccine efficacy depends on strain cross-protection and effective vaccination program implementation. We evaluated a multi-faceted IAV vaccination strategy which included (a) monthly surveillance of pigs at weaning, (b) selection of epidemiologically relevant strains from farms under surveillance, (c) updating IAV strains in custom-made vaccines, and (d) seasonal mass vaccination with custom-made vaccines given to sows in 35 farrow-to-wean farms within an integrated swine farm system. Reduction of IAV in pigs from vaccinated sows was determined by monthly monitoring of farms for 30 months by IAV rRT-PCR (PCR) testing of nasal wipes collected from litters of piglets at weaning. Hemagglutinin (HA) nucleotide and amino acid (AA) sequence homology of the circulating and vaccine strains was determined by pairwise alignment and AA comparison at antigenic sites. Of the 35 farms monitored, 28 (80%) tested positive at least once, and 481 (5.75%) of 8352 PCR tests were IAV positive. Complete HA sequences were obtained from 54 H1 (22 H1-δ_1B.2.1, 28 H1-γ_1A.3.3.3, and 4 H1-pdm_1A.3.3.2 clades) and 14 H3 (12 IV-A 3.1990.4.1 and 2 IV-B 3.1990.4.2 clades) circulating IAV strains. During the study, custom-made vaccines were updated three times (eight strains total) and administered to sows at five distinct time periods. The HA AA similarity between vaccine and circulating strains ranged from 95% to 99%; however, the 0 to 71% similarity at HA antigenic sites prompted the vaccine updates. Herd IAV prevalence decreased from 40% (14/35) to 2.9% (1/35), accompanied by a numerical reduction in IAV-positive samples post-vaccination. Our results support having a comprehensive approach to controlling influenza in swine herds that includes surveillance, vaccination, and careful program implementation to reduce IAV in pigs.

## 1. Introduction

Respiratory disease caused by influenza A virus (IAV) infection has a negative effect on pig health and productivity [1] and is a public health concern because novel reassortant viruses of zoonotic potential may emerge from pigs [2,3]. IAV-infected pigs tend to be more susceptible to secondary bacterial infections [1], which results in increased antimicrobial use and increased production costs [3]. The impact of IAV on swine productivity, the public health implications, and the desire to reduce antibiotic usage in pigs are factors that drive the development of strategies to control and ideally eliminate IAV from swine herds.

Vaccination is the most common tool used to control IAV in pigs [4]. Approximately 50% of large farms report using IAV vaccines in breeding farms in the United States, with sow vaccination being the most common protocol [5]. Sow vaccination prior to farrowing is generally conducted between 80 and 100 days of gestation in an attempt to increase the transfer of maternally derived antibodies (MDAs) to piglets [6] and reduce the clinical impact of the disease. Both commercial vaccines and custom-made, licensed, autogenous vaccines are used in North American swine farms. Commercial vaccines tend to include nationally relevant IAV strains selected to protect against predominant strains [7] but may not provide cross-protection to specific strains circulating within a given herd. In contrast, custom-made vaccines, i.e., autogenous biologics, which require a valid veterinary client patient relationship, can be updated regularly and are developed using IAV strains that are circulating within a herd or an epidemiological unit [8]. However, licensing of custom-made vaccines requires demonstration of purity but not potency or efficacy data [8].

In the US, custom-made vaccines account for approximately 80% of the influenza vaccines used in large swine-breeding farms [5]. The high genetic diversity of IAV strains from pigs [9] and the presence of many antigenically distinct clades [10] make control of influenza in pig farms very challenging [11]. In general IAV vaccines are used to control influenza clinical disease [12] and can be effective at reducing IAV prevalence [13]. Both custom-made and commercial vaccines administered either pre-farrowing or through whole herd mass vaccination have been reported to reduce IAV prevalence at weaning [14]. Furthermore, IAV vaccines have also been used in efforts to eliminate influenza from farms [15]. However, the factors that influence the success of IAV vaccination programs in swine herds are not well understood, including the type of vaccine (commercial vs. custom-made) and administration protocols (e.g., whole herd/mass vaccination vs. pre-farrowing). An H3N2-based custom-made vaccine administered following a whole herd administration protocol was reported to result in the elimination of IAV from a breeding herd [9]. In contrast, a commercially available vaccine that contained strains that were genetically distinct from the H3N2 strain circulating in a farm was reported as less efficacious in a separate study [15].

Due to the remarkable genetic and antigenic diversity of IAV, vaccine strain selection is considered a key determinant of vaccine efficacy [16]. Vaccine strain selection depends, in part, on the antigenic matching [16] of the vaccine virus with the wild-type virus(es) infecting the pigs. Given the IAV strain variability [8] and the frequent introduction of new IAV strains into pig farms [17,18], the suitability of vaccine strains needs to be regularly reassessed. In humans, influenza vaccine composition is determined by annual selection of strains to update the vaccines and immunize populations before the epidemic season [16]. Influenza vaccine selection is determined by a scientific committee convened by the World Health Organization (WHO) [19]. The vaccine selection committee considers a combination of surveillance (Global Influenza Surveillance and Response System—GISRS), data analysis, coordination between human and animal influenza networks, antigenic and genetic characterization of viruses, epidemiological surveys, mathematical models, and laboratory assays (most commonly the hemagglutination inhibition (HI) assay) [19]. This coordinated effort results in human seasonal vaccines being updated twice yearly to minimize mismatches between vaccine strains and circulating strains [16] that may occur in the influenza season of each hemisphere. In contrast, the selection and design process for influenza vaccines for pigs does not follow this systematic global approach, and decisions must be made at the farm, system, or regional level. Regardless of whether the vaccine is for humans or animals, the hemagglutinin (HA) protein is extensively evaluated. Data from HA protein evaluations provide information regarding the potential protective immunity conferred by the vaccine since HA is the target of protective immune responses and the major component in vaccine production [20]. Within the HA protein, certain amino acid positions are associated with antigenic response and receptor binding sites. These amino acids are positioned at key antigenic sites and have been described for both H1 [21] and H3 [22] IAVs. Changes in amino acids at key antigenic sites can alter vaccine efficacy due to changes in antigenic cross-reactivity (vaccine mismatch) [23].

In addition to the viral and host factors that can contribute to vaccine failure, farm management factors and the complexity of swine production systems can accentuate the problems that result in vaccine failure. Therefore, veterinarians face difficult decisions when implementing programs to control influenza, and the decisions are often made with limited information and resources while also being constrained by other health or production priorities. Commonly, swine veterinarians base their decisions to use a specific influenza vaccine on cost and availability and then evaluate the effectiveness of that decision by measuring clinical signs and/or production parameters post-vaccination [4].

To advance the IAV vaccine decision-making process, we conducted a collaborative farm study to document and evaluate a multi-year IAV control strategy to reduce IAV in pigs at weaning in the breeding farms of an integrated swine farm system owned by a single company. The strategy included routine, targeted surveillance of IAV, identification of epidemiologically relevant IAV strains in the farms, surveillance-based updates of the IAV strains in the company’s custom-made vaccines and implementing seasonal mass vaccination twice a year in breeding herds using the selected vaccines.

## 2. Materials and Methods

### 2.1. Farm Selection

Thirty-five farrow-to-wean farms belonging to an integrated swine farm system in the Southeastern United States developed and implemented an IAV surveillance program in pigs of weaning age, e.g., 18 to 28 days of age. Data on detection of IAV was collected for 2.5 years, from July 2016 to January 2019, and farms were classified as IAV positive or negative based on diagnostic results obtained from the surveillance program. Farms had a history of IAV vaccination using a quadrivalent, oil-in-water emulsion adjuvanted, whole inactivated vaccine readily available commercially in the United States. The farms had a mean population of 3000 sows (range of 1500–5000), and all 36 farms were located within the same state and followed common industry production management practices [1], with production parameters similar to published benchmarks [24]. Over 65% of the farms were porcine reproductive and respiratory syndrome virus (PRRSV) stable [25] and *Mycoplasma hyopneumoniae* positive. Occurrence of IAV in the breeding herds was determined by monthly surveillance of 10–12 litters prior to weaning (described below in sample collection and processing) using nasal wipes, with some farms having more than one sampling event per month. A sampling event was described as a submission of nasal wipes collected from a single farm at a single time point. Nasal wipes were tested for IAV by a reverse transcription real-time polymerase chain reaction (rRT-PCR) test in veterinary diagnostic laboratories in the United States that were American Association of Veterinary Laboratory Diagnosticians accredited. An IAV-positive sampling event was categorized as having at least one nasal wipe collected with an IAV rRT-PCR-positive Ct value of ≤37.5, i.e., a positive result. Accordingly, IAV negative sampling events are those with all nasal wipes collected having no Ct or a Ct value > 37.5, i.e., a negative result.

All breeding herds received replacement gilts from the integrated swine farm system’s gilt multiplication farms. The gilt multiplication farms had low to no detectable levels of IAV based on monthly IAV monitoring by rRT-PCR. Replacement gilts were vaccinated with an IAV custom-made vaccine at ~20–22 weeks of age and then revaccinated upon arrival at the breeding herd at about 24–25 weeks of age. The vaccines used in the gilts were the custom-made vaccine serials available to the farms at the time of gilt introduction (described below in vaccine manufacturing and administration and in Table 1). Gilt movements varied between farms, based on the acclimatization needs, management, and presence of an external quarantine site to acclimatize gilts before introduction into the breeding herd.

### 2.2. Sample Collection and Processing

Throughout the study period, there were two similar, yet distinct sample collection protocols. From July 2016 to February 2018, three to five pigs from the same litter were sampled using the same nasal wipe for a total of 10 litters in each farm. The 10 nasal wipes were collected monthly and tested individually using an IAV rRT-PCR [26]. From March 2018 to January 2019, the same sampling strategy was followed, but samples were collected from twelve litters which were tested using the same rRT-PCR procedures but on pools of three wipes per test. In summary, the protocols differed in the number of rRT-PCR tests conducted (ten vs. four) and the number of pigs sampled (30 to 50 piglets total from 10 litters vs. 36 to 60 piglets from 12 litters). This change in protocol was undertaken to save costs yet maintain diagnostic sensitivity [27]

Nasal wipes were collected by wiping the exterior of the pig snout using a 4 × 4 sterile gauze wetted with a liquid solution to collect the pigs’ nasal and oral secretions. At the beginning of the study, the liquid solution included 10 mL of phosphate-buffered saline (PBS) by Gibco™ (Grand Island, NY, USA). In November 2018, it was changed to DMEM, Dulbecco’s Modified Eagle Medium Gibco™ (Grand Island, NY, USA), supplemented with antibiotics and antimycotics to increase the likelihood of IAV isolation in cell culture [27]. After collection, all the samples were placed in insulated containers with ice gel packs and then transported to the veterinary diagnostic laboratory for IAV rRT-PCR testing.

### 2.3. Diagnostic Tests

#### 2.3.1. Influenza A Virus rRT-PCR

Viral RNA was extracted from the nasal wipes using a magnetic particle processor procedure (Ambion^®^ MagMAX™AM1835, Viral RNA Isolation Kit; Applied Biosystems, Foster City, CA, USA). Extracted RNA was tested by rRT-PCR to detect the IAV matrix gene [28]. Samples having rRT-PCR results with cycle threshold (Ct) values ≤ 37.5 were considered positive and Ct > 37.5 were negative [14].

#### 2.3.2. Cell Culture for Influenza A Virus Isolation

At least one rRT-PCR positive nasal wipe with the lowest Ct value from each farm was used to perform virus isolation using Madin–Darby canine kidney (MDCK) cells [29,30]. MDCK cells were prepared in 6-well plates for each selected sample. Wells were inoculated with 200 µL and 100 µL of sample, in duplicate, and incubated for 1 h at 37 °C with 5% CO_2_. A volume of 1.5 mL of DMEM (Gibco™, Grand Island, NY, USA) was supplemented with 7.5% bovine serum albumin (Gibco™, Grand Island, NY, USA), 1X antibiotics and antimycotics (Gibco™, Grand Island, NY, USA), 750 µL 1 mg/mL trypsin-TPCK, gentamicin, and neomycin, and then the plates were incubated at 37 °C with 5% CO_2_. Plates were evaluated at day 3 and 5 for appearance of a positive cytopathic effect (CPE). All the wells with positive CPE were confirmed as IAV by a hemagglutination assay using 0.5% turkey red blood cells and VetScan Avian Influenza Type A Virus Rapid Test (Alere Scarborough Inc., Union City, CA, USA).

#### 2.3.3. Viral Genetic Sequencing

The IAV hemagglutinin (HA) gene of the viral isolates was sequenced as previously described. Briefly, viral RNA was obtained from the virus isolate with a QIAamp RNeasy Mini Kit (QIAGEN, Inc., Valencia, CA, USA), using the protocol recommended by the manufacturer [28]. Sequencing primers were used based on a previous report [30]. For HA, rRT-PCR products were then purified using a QIAamp Gel extraction kit (QIAGEN) and then submitted for Sanger sequencing using a fully automated ABI 3730xl DNA Analyzer (Perkin-Elmer SeqGen, Inc. Torrance, CA, USA) with ABI BigDye Terminator version 3.1 chemistry—Perkin-Elmer (Applied Biosystems 2002). Once HA sequences were obtained, these were aligned and assembled using the Clustal W algorithm [31] in Geneious 11.0.4 (http://www.geneious.com) [32].

### 2.4. Custom Vaccine Strain Selection and Sequence Analysis

Using the software suite in Geneious 11.0.4 [32], the nucleotide sequences were aligned with published reference sequences [7,33] using the Clustal W alignment algorithm (http://www.ebi.ac.uk/tools/clustalw2, accessed on 29 January 2025), and phylogenetic trees were made using the neighbor-joining clustering method to determine relationships between strains. Obtained nucleotide sequences were translated into amino acid sequences. Amino acid sequences were aligned using Clustal W (http://www.ebi.ac.uk/tools/clustalw2, accessed on 29 January 2025) and Jukes–Cantor models were used to calculate distance (number of differences in amino acids between strains). The percent identity provided by the Jukes–Cantor models was used to determine the similarities of the viral strains isolated from the IAV-positive farms and their percent amino acid identity to the custom-made vaccines used prior to and/or at that time point in each farm. Genetic analyses of the HA genes and proteins of the herd IAV strains and vaccine IAV strains were performed every two months throughout the 2.5 years of the study.

The vaccine selection criteria to update the custom-made vaccines were similar to those recommended by Sandbulte [7]. Briefly, the overall percent HA protein amino acid identity between newly detected herd IAV strains and strains previously included in the custom-made vaccines was calculated. Additionally, specific amino acids in the HA protein of the evaluated strains were compared to determine if there were any differences at specific amino acid sites that would potentially be correlated with a change in antigenicity. Specifically, we compared the following amino acids, as listed below, with all numbers listed based on mature protein numbering:For H3 strains: the 7 key amino acids at positions 144, 155, 156, 158, 159, 189, and 193 [22];For H1 strains of swine 1A lineage (alpha, beta, gamma, and pandemic), the 51 amino acids are listed in quotation marks at these antigenic sites: Sa “124,125,153,154,155, 156, 157, 159, 160, 161,162, 163, and 164”; Sb “184, 185, 186, 187, 188, 189, 190, 191, 192, 193, 194, 195, and 196”; Ca1 “166, 67, 168, 169, 202, 203, 204, 205, 235, 236, and 237”; Ca2 “137,138, 139, 140, 141, 142, 143, 221, and 222”; and Cb “70, 71, 72, 73, 74, and 75” [21];For H1 strains of human 1B lineage (delta 1a, delta 1b, and delta 2), the 23 putative key amino acids, specifically those at sites 69, 119, 121, 127, 129, 133, 140, 152, 167, 174, 185, 188, 208, 214, 215, 221, 255, 258, 269, 272, 288, 307, and 309 [19].

Not only were sequences analyzed to determine the percent HA amino acid similarity and comparisons of amino acids at specific sites, but also all IAV strains detected were subtyped and clade-classified using the NIAID Influenza Research Database (IRD) [33] and octoFLU tools [34,35]. Newly identified strains were considered vaccine candidates if the strains possessed these characteristics: (1) evidence for increasing dominance of the newly identified strains as determined by frequent detection in sampling events and patterns observed in the phylogenetic tree; (2) having substantial genetic differences from the custom-made vaccine strain, specifically sharing <95% HA protein amino acid overall identity between newly identified dominant strains and the custom-made vaccine strains; and (3) possessing amino acids in the HA antigenic sites described above that differed from the custom-made vaccine strains. The vaccine candidate strains were subjected to final selection after discussions with the farm system’s veterinarian regarding the strain’s clinical impact or clinical relevance (for example, was the strain causing clinical disease in the herd and/or was the strain likely to be disseminated by multiple other herds due to pig movements). After selecting the vaccine candidate strains, the viral isolates of the IAV strains were submitted to a commercial vaccine manufacturer to prepare a custom-made inactivated vaccine product. The vaccine was updated and the vaccine containing the new strain(s) was administered.

### 2.5. Vaccine Manufacturing and Administration

At the request of the farm system’s veterinarian, IAV vaccines were prepared by an animal pharmaceutical company according to USDA autogenous vaccine regulations [29]. The IAV isolates were submitted to the vaccine company, cultured on MDCK cells, inactivated after culture, and the killed antigen content (antigenic mass) [7] in the final product was placed in bottles according to farm requirements. Emulsigen^®^-D adjuvant (MVP, Phibro Animal Health Corporation, Teaneck, NJ 07666, USA) was utilized to adjuvant the inactivated influenza viruses. Custom-made vaccines were administered seasonally in the fall (September, October, November) and spring (March, April, May) to the entire breeding herd in every farm using a mass vaccination approach wherein all dams received a single intramuscular 2-mL dose of the custom-made vaccines.

### 2.6. Data Analysis

The proportion of positive sampling events was calculated by dividing the number of positive sampling events by the total sampling events in the study. A farm was considered positive if any sampling event was positive. Monthly farm-level IAV prevalence was calculated by dividing the number of positive herds by the total number of herds tested in a given month. Herds that did not use vaccination at each vaccination point were also included in the analysis. Finally, a generalized linear mixed model (GLMM) was fitted using the number of positive and negative samples as a binomial outcome to estimate the mean proportion, with 95% confidence intervals, of IAV positive samples for each vaccine administration period [36]. An interaction term between the sampling point (three months before or after vaccination) and vaccination (yes/no regardless of the specific vaccine used) was added as a predictor. We used data from the three consecutive months before and after vaccine administration rather than one month because three months is more representative of the impact of vaccine-derived population immunity [7,37], better represents the lasting effect of vaccination within a season, and allows a defined epidemiological analysis based on pig production cycles. The random effect of a farm was included in the model to account for the lack of independence between observations from the same farms.

## 3. Results

### 3.1. Summary of IAV rRT-PCR Results and Prevalence Calculations

A total of 131 out of 778 (16.8%) sampling events between July 2016 and January 2019 were considered IAV positive. Out of 8352 IAV rRT-PCR tests (on nasal wipes tested individually or in pools), 481 (5.75%) were rRT-PCR positive for IAV, and 68 complete hemagglutinin sequences were obtained. Out of 35 farms enrolled in the study, 30 (86%) tested IAV positive at least once during the study period (Figure 1). Farm-level prevalence declined progressively from 14.3 to 31.4% in sampling events conducted in 2016 to 2.9% in January 2019.

### 3.2. Custom-Made Vaccine Composition and Updates

Custom-made vaccines were updated three times throughout the study period (Table 1). Vaccine A (code 1317) was a trivalent vaccine that included strains which were considered representative of the dominant circulating strains as of June 2016, specifically strain 1A, an H1N1 gamma of clade 1A.3.3.3; strain 2A, and an H1N2 delta 2 of clade1B.2.1; and strain 3A, an H3N2 IVA of clade 3.1990.4.1 (Appendix A, H3 phylogenetic tree). In March 2017, an H1N1 pdm of clade 1A.3.3.2 (Appendix A, H1 phylogenetic tree) was identified in two herds but was not included in the vaccine because the veterinarian considered it had a minimal clinical impact on the pig population. Later in the same year, July 2017, an H1 delta2 of clade 1B.2.1 IAV (strain 1B) was identified; strain 1B shared <93% identity with the H1 delta 2 clade 1B.2.1 IAV strain 2A contained in vaccine A. Therefore, vaccine B (code 1350) was formulated to include this new H1 delta2 of clade 1B.2.1 circulating (strain 1B), which replaced the H1 delta2 of clade 1B.2.1 (strain 2A) contained in vaccine A. Selecting strain 1B as a vaccine candidate for the control of IAV caused by H1 delta2 of clade 1B.2.1 infections in the herds was supported by analyses that showed strain 1B shared >97% similarity and had identical amino acid profiles to the H1 delta2 clade 1B.2.1 strains most frequently detected throughout the farm system between July 2017 and September 2017. Concurrently, the H3 CI-IVA of clade 3.1990.4.1 (strain 3A) in vaccine A was replaced due to the frequent detection of two subclusters of H3 strains that were distinct, i.e., <95% HA gene nucleotide and protein amino acid identity, from H3 vaccine strain 3A and antigenically different at key antigenic sites. In detail, the most frequently detected H3 Cl-IVA had an amino acid motif of KYNNYKY, and H3 vaccine strain 3A had an amino acid motif of NYKNYSS, such that there were differences at four of the seven key amino acid sites, namely K145N, N156K, K189S, and Y193S. In summary, vaccine B (code 1350 NEW) was a bivalent vaccine containing H1N2 delta2 clade 1B.2.1 strain 1B and H3N2 IVA clade 3.1990.4.1 strain 2B.

In November 2017, an H1N1 gamma circulating strain was identified as a suitable vaccine candidate to match the frequently detected gamma H1N1 clade 1A.3.3.3 farm strains, to which the gamma vaccine candidate had a percentage identity above 95% to the majority of gamma farm strains. More specifically, the HA protein amino acid similarities between circulating gamma H1N1 clade 1A.3.3.3 strains and the candidate gamma H1N1 clade 1A.3.3.3 vaccine strain ranged from 96.3% to 99.8%. Since the goal was to select a vaccine strain that shared >95% overall HA amino acid similarity with the circulating gamma clade 1A.3.3.3 strains identified between July 2016 and November 2017, H1N1 gamma clade 1A.3.3.3 (strain 1C) was included with the H1N2 delta2 clade 1B.2.1 strain 1B and H3N2 IVA clade 3.1990.4.1 strain 2B to formulate the inactivated trivalent vaccine C (code 1379) used in 2018 (Figure 2).

### 3.3. Effect of Vaccination on IAV Occurrence

Herd IAV status changed over time (Figure 1). During the 3 months prior to the first vaccine administration, sixteen herds were identified as IAV positive. After vaccination, fourteen herds were positive. Thirteen and seven herds were identified as positive before and after the second vaccine administration, respectively. Before the third vaccination, the same seven herds remained positive but one additional herd tested positive after the administration of the vaccine. Before the fourth vaccination, two different herds tested positive, and the same number of herds were identified as positive after the vaccine administration. Finally, two herds were positive before the fifth vaccine administration.

The proportion of IAV-positive nasal wipes obtained during the three consecutive months before and after each vaccination changed throughout the time period of the study (Table 2). The first vaccine (vaccine A) was administered in the fall of 2016 to 28 of the 35 enrolled farms. During the three months (September, October, and November 2016) before vaccine A was administered in December 2016, there were 170 out of 1050 individual samples (16.2%, CI: 14.1–18.5%) that tested IAV positive. In comparison, during the three months (January, February, and March 2017) after vaccination, there were 125 out of 965 (13.0%, CI: 10.9–13%) positive samples, which was a 3.2% reduction in the proportion of samples testing IAV positive. In the spring of 2017, vaccine A was administrated again to 32 of the 35 herds in the farm system. In the pre-vaccination period (February, March, and April 2017), 114 (11.7%, CI: 9.8–13.8%) of 978 collected samples from all the herds, tested IAV positive. In the three months post-vaccination (June, July, and August 2017), less samples collected tested IAV positive (26 of 963 (2.7%, CI: 1.8–3.9%)).

In fall 2017, 31 of the 35 herds received vaccine B, two herds received vaccine A again, and three herds did not receive any vaccine. In the three months pre-vaccination (July, August, and September 2017) with vaccine B, 30 (3.1%, CI: 2.2–4.4%) of 975 samples tested IAV positive. Three months post-vaccination (November 2017, December 2017, and January 2018), 67 (6.7%, CI: 5.2–8.3%) of 1004 samples tested IAV positive. In the spring of 2018, a fourth vaccination protocol was implemented, where the animals in the herds were given one of the three vaccines—A, B or C—with their choice based on vaccine stock availability and farm staff/veterinary preference. For example, farm 7 detected a circulating delta2 H1 strain with 96% HA amino acid similarity to the delta2 contained in vaccine B (strain code 1B) and decided to continue using vaccine B. However, a new circulating gamma strain that was identified and included in vaccine C was used in farms 20 and 24. Out of all 35 farms, in the three months (December 2017, January 2018, and February 2018) pre-vaccination with vaccines A, B, or C, 30 samples were IAV positive (3.1%, CI: 2.2–4.4%) of the 975 samples collected. In the three months post-vaccination (April, May, and Jun 2018), 1221 samples were collected and tested for IAV by rRT-PCR in pools of three. Of the 407 pools, nine pools (2.2%, CI: 1.2–4.2%) were IAV positive.

Finally, a fifth vaccine administration with vaccine C occurred in the early fall of 2018. In the three months (May, June, and July 2018) pre-vaccination, 1197 samples were tested by rRT-PCR in pools of three (399 pools) and only three (0.8%, CI: 0.3–2.2%) pools were positive. In the three months post-vaccination (September, October, and November 2018), 1263 samples tested in pools of three (421 pools) yielded 8 (1.9%, CI: 1–3.7%) positive pools.

In addition to a change (usually a decrease) in the number of IAV-positive samples post-vaccination, there was also an effect of vaccine administration on the estimated proportion of positive IAV samples (Table 3). The generalized linear mixed model revealed that samples collected before vaccination had a significantly higher probability of testing positive compared with those collected after vaccination for all vaccination periods analyzed. Results before and after the fourth vaccination could not be modeled given the change of sampling strategy (samples tested individually in some months then in pools other months). The interaction term between the sampling point and vaccination was a statistically significant predictor in the GLMM model. Farms that did not use vaccines had a statistically significant increase in the proportion of IAV-positive samples, while farms that vaccinated had a statistically significant reduction in the proportion of IAV-positive samples. However, the between-herd variance in the GLMM model was 4.6, indicating differences in the change of positive proportions in the months before and after vaccination were attributable to the vaccination and less likely due to a particular characteristic of the herd.

### 3.4. Analyses of Hemagglutinin Protein Amino Acid Sequences and Amino Acids at Antigenic Sites

Of the 68 complete HA sequences obtained, there were 54 H1 (22 H1N2-δ2 clade 1B.2.1, 28 H1N1-γ clade 1A.3.3.3, and 4 H1N1-npdm clade 1A.3.3.2) and 14 H3N2 (12 IV-A clade 3.1990.4.1 and 2 IV-B clade 3.1990.4.2) circulating IAV strains identified. The overall HA amino acid similarity between the strains of the same clade contained in the vaccines and the strains circulating in the pigs ranged from 80% to 100% (Table 4). However, identity at HA antigenic sites measured by the concordance of amino acids between herds and vaccine strains ranged from 0% to 71% (Table 5). For the HA amino acid site comparisons, strains usually differed at several amino acid sites and not just one or two. However, there were certain amino acid sites that were more likely to have a difference than others. For example, for the H3 strains, the most common amino acid difference noted was N145K with 80% (8/10) of the H3 circulating strains and the H3 vaccine strains differing at this site. For H1 gamma clade 1A.3.3.3 strains, the most commonly noted amino acid difference between vaccine strains and circulating strains was K142N (44%, (12/27)). For the H1 delta clade 1B.2.1 strains, almost 90% (24/27) of the circulating strains had V152E compared to the H1 delta clade 1B.2.1 vaccine strain (Table 6). The maximum number of amino acid differences within the seven antigenic sites of H3 viruses was five. On the other hand, we found that the maximum amino acid changes in gamma clade 1A.3.3.3 and delta clade 1B.2.1 H1 strains were 12 of 51 and 14 of 23 amino acids, respectively.

## 4. Discussion

Vaccination is the most effective strategy to protect humans from influenza infections [38] and the most common tool to control influenza in pig populations [39]. In pigs, homology between circulating (herd) and vaccine strains is considered important [12] since higher homology tends to be correlated with a higher probability of disease control through reduced clinical disease [7], lower risk of transmission [13], and decreased disease prevalence [14]. In this study, we report the results of a system-wide influenza vaccination control strategy to reduce IAV in piglets at weaning. The strategy was implemented by an integrated farm production system willing to allow collaborative research and commit to a comprehensive program that included surveillance, analysis of the HA amino acids of strains circulating in the farms to assist the vaccine strain selection process, and frequent update of the custom-made vaccines to include the selected strains. Our results from a two and one-half years analysis using three distinct vaccines administered at five different time periods indicated a reduction in influenza positivity due to vaccination in three of the five vaccine administration periods according to the generalized linear mixed model. Overall, farm-level prevalence of IAV in weaned pigs declined from 40% in July 2016 to 2.9% in December 2018. Our results suggest that mass vaccination using custom vaccines that incorporate clinically relevant herd strains as part of a comprehensive surveillance and control plan has the potential to reduce influenza in weaned pigs.

We evaluated the impact of vaccination on IAV prevalence by comparing the proportions of positive samples pre- and post-vaccination. We decided to conduct the analysis with the data from three months before and after vaccination to better estimate the lasting effect of vaccination, including the transfer of maternal immunity to piglets, given that duration of IAV immunity post-vaccination is estimated to last between 56 and 112 days [40] and that it usually takes a minimum of 15 to 21 days for the protective response to be noted in pigs that nurse vaccinated sows [41]. A three-month period also reflects better the expectation of producers to benefit from the investment made on vaccination. In addition, three months almost represents the length of a gestation period (114 days on average), with the resulting immune status of the progeny population.

Vaccine A was administered three times (the time periods of fall 2016 and spring 2017). Even though there was a reduction in the percentage of positive samples in both time periods, the most consistent reduction in the proportion of positive samples happened after the second administration (from 11.7% to 2.7%). We hypothesize that this reduction was due to acquiring a much broader herd immunity after the second vaccination in this farm system that had a history of exposure to both natural IAV infections and vaccinations while also coinciding with the seasonal IAV reduction expected in the spring months of April, May, and June. This seasonal pattern has been already documented in pigs [42] and humans [43,44]. For pigs, at the herd level in breed-to-wean farms, the seasonal reduction was associated with absolute humidity in outdoor air and temperature [2]. In contrast, after use of vaccine B, there was an increase in the proportion of positive samples from 3.1% to 6.7%. We suggest that the increase in prevalence may have been due to the circulation and emergence of a new strain (strain 1C) in the population. To overcome this mismatch and to decrease the apparently rising rate of IAV positivity, the addition of the new circulating strain (1C) into the updated vaccine C was made. In addition, during the fourth vaccination there was flexible use of either one of the three vaccines (A, B, and C) including those that were used in the past and matched better with the circulating strains at each farm at that time, resulting in an appreciable reduction in the proportion of positive samples from 5.1% to 2.2%.

The results of the generalized linear mixed model generally paralleled the trends observed in the descriptive analysis with a few exceptions. After the first, second, and third vaccinations, there were reductions in the proportion of positive samples. The analysis could not be modeled for any of the vaccines used on the fourth vaccination due to the change of sampling strategy before (individual samples) and after (samples tested in pools), and, therefore, there was not a feature-by-feature comparison. Overall, we saw a reduction in the occurrence of IAV detection in pigs at weaning when the vaccine strain was updated to include the most commonly detected circulating IAV strain(s) except after administration of the third vaccination.

The selection of strains for the production of custom vaccines is complex. First, it requires detection of the circulating strains at the herd level through a systematic surveillance program. Second, it requires characterization of the strains to identify which ones are not only epidemiologically relevant but possess the antigenic characteristics capable of inducing broad cross-protection against the circulating strains. We compared strains genetically and antigenically, not only considering the complete hemagglutinin protein similarity but also evaluating key amino acids at specific antigenic sites considered significant for protection. The hemagglutinin (HA) protein was the focus of the genetic and antigenic analyses since HA is responsible for the escape of the influenza A virus from pre-existing immunity [45,46]. Even though the analyses were HA focused, the analyses still required qualified and trained technical personnel capable of performing computational analyses. In addition, optimizing the surveillance and testing programs (sample size) to identify the epidemiologically relevant strains that were present in several breeding herds repeatedly and causing clinical signs of disease, as well as advancing understanding of the immunological implications of amino acid changes at key antigenic sites are necessary to refine this approach.

Studies in humans have demonstrated that the distance between each antigenic cluster is sufficient to require an update in influenza vaccines [22]. Sixty-seven amino acid positions in the HA protein have been associated with the divergence of one cluster to another, which is considered sufficient to require an update in influenza human vaccines with H3N2 influenza virus [22]. In this study, nucleotide and amino acid similarities were calculated between all strains contained in the administered vaccines. Most of the comparisons indicated the HA protein amino acid sequence similarity was over 95%, suggesting that the current vaccine strain in use would likely provide immunity against the herd strains detected and circulating. However, herd IAV strains with HA protein amino acid similarities <95% to the vaccine strain in use or available were also detected, and it was the detection of these divergent strains that prompted vaccination updates. These changing or divergent IAV strains were identified and considered potential vaccine candidates if they continued to be detected at an increasing rate in the farms in subsequent months and were considered clinically relevant by the veterinarian.

Sequence and antigenic site analysis also further defined the identification and discernment of differences between circulating and vaccine strains. Amino acid substitutions that alter antigenic sites can inhibit antibody binding and allow the virus to escape immunity [47]. In our analysis, N145K was the most common site where differences were found between herd circulating H3 strains and H3 vaccine strains. This was also the most frequently identified difference in 445 of 1007 H3N2 viruses evaluated in swine in the United States from 2012 to 2016 [48]. Previous research has demonstrated that six to seven amino acid changes in the HA protein are enough for marked antigenic differences in human and swine H3 viruses [47]. However, our analysis suggests that five amino acid changes in antigenic sites of H3 circulating strains may warrant an update in vaccine strains. Based on this information, we could infer that those differences among the herd-circulating strains were resulting in antigenic differences that could elude immunity. On the other hand, H1 strains had different patterns of amino acid changes than H3 viruses. Several amino acid differences were identified when comparing circulating and vaccine strains, and upon first glance, these differences appear to have no appreciable trend. However, the K142N change at the Ca2 epitope site of the H1 gamma strains, the most commonly detected change in our amino acid comparisons, was present in 44% of the herd strains evaluated, and has been described as antigenically important [49]. An amino acid property change at this site changes the IAV receptor on the host cell, which then alters the attachment of the antigen with a subsequent change in antibody response [49]. As with H3 amino acid evaluations which focus on position 145 as key to antigenicity, as demonstrated by Abente et al. [22], our results suggest that for H1 gamma amino acid evaluations, attention should be paid to position 142 because an amino acid switch from K to N at position 142 results in a change in amino acid properties that alters receptor binding [49].

Having a surveillance program is crucial when designing an effective IAV control program. However, a comprehensive surveillance program can be costly and difficult to implement consistently in an integrated farm system with competing priorities and limited resources. The number of samples collected at each sampling event was also increased during the study from 10 to 12 nasal wipes with the caveat that pooling of three samples in one rRT-PCR reaction was incorporated when that change was made. However, pooling in three had a sensitivity of 94% [26], and this would not account for observed differences. Therefore, we do not believe these changes impacted significantly the results of the study since pooling up to five samples has not resulted in a significant decrease in the rRT-PCR sensitivity [27] and the farm system did assess the effect of pooling in the probability of detecting a positive sample in a subset of samples prior to making a system-wide change with favorable results [personal communication, results not shown]. Furthermore, the changes made represented significant diagnostic cost savings as each sampling event decreased from USD $310 to USD $124 just in diagnostic costs. Decreasing the cost per sampling event was beneficial to maintaining the surveillance under commercial industry conditions.

## 5. Conclusions

In this study, we have reported the results obtained after the implementation of a farm system-wide strategy to control influenza. At the termination of this study, herd prevalence was 2.9% compared with the initial prevalence of 40%. The results were remarkable as a whole, but the farming system recognizes the work that still needs to be carried out in order to fully reap the benefits of having an influenza control program in place. Although the program was centered on surveillance and the identification of strains to produce a custom-made vaccine and its timely administration, other factors may have also contributed to the overall reduction of IAV infections. These factors included the introduction of IAV-negative gilts into the herds, increasing personnel knowledge of influenza through training and education, farm worker vaccination to decrease human-to-swine transmission, and improvement of internal biosecurity measures, all of which may have collectively reduced the introduction of new IAV strains and decreased the transmission of endemic strains. As such, we cannot attribute the reduction in prevalence solely to the vaccination strategy. Nevertheless, sow vaccination has been shown to be one of the most important factors in reducing IAV occurrence in pigs prior to weaning [14], and our observations in this large farm setting are consistent with those studies. Our study also highlighted the difficulty of implementing such a comprehensive program across multiple farms in an integrated farming system, in particular for the veterinarian in charge of the health program because of the technical and organizational issues. These limitations are even more significant in countries where more advanced diagnostic technologies are not available.

This study provides new information that shows possible ways to overcome the difficulties of selecting strains for inclusion in custom-made IAV vaccines to control influenza. We demonstrated that a surveillance program and the implementation of a vaccination program in an integrated swine farming system can play a role in reducing the prevalence of IAV in swine farms. To our knowledge, this is the first study that describes the effect of the implementation of a system-wide surveillance and vaccination program in a large-scale, integrated swine farming system over multiple influenza seasons and highlights challenges that the swine industry continues to face regarding influenza. Despite the challenges in implementing such a program, this extensive, collaborative work indicates that taking a comprehensive approach to control influenza in swine breeding herds can result in the overall reduction of IAV in pigs at weaning. While the economic benefit of this approach needs to be further assessed, particularly with respect to the health and productivity outcomes in the weaned pigs as they grow to market size at farms across the system, the positive outcomes of vaccination presented here should be considered by swine, poultry, and other livestock farmers and veterinarians when deciding whether or not to include vaccination as part an overall IAV control program. Comprehensive IAV surveillance and vaccination is possible and can benefit both animal and human health, animal welfare, and food security.

## Figures and Tables

**Figure 1 viruses-17-00240-f001:**
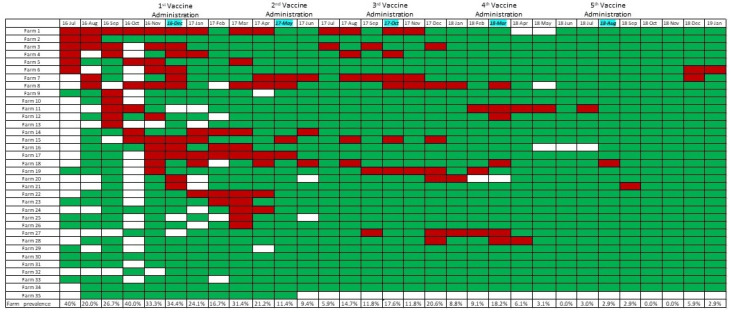
Influenza A virus (IAV) farm status assessed by IAV rRT-PCR test results per farm (rows) by month (columns) from July 2016 to January 2019. A farm was considered to be of positive status during that month if at least one sampling event was positive. Red cells indicate positive results for the farm, and the farm is given an influenza positive status for that month. Green cells indicate negative results for the farm that month, and the farm is given influenza negative status; white cells indicate no samples were collected from that farm during that month. Months (columns) in bold italics with cyan highlighting indicate vaccination administration times. Farm prevalence was calculated by dividing the number of positive herds (if at least one sample tested positive) by the total number of sampled herds in a month and is given at the bottom of each column for each month.

**Figure 2 viruses-17-00240-f002:**
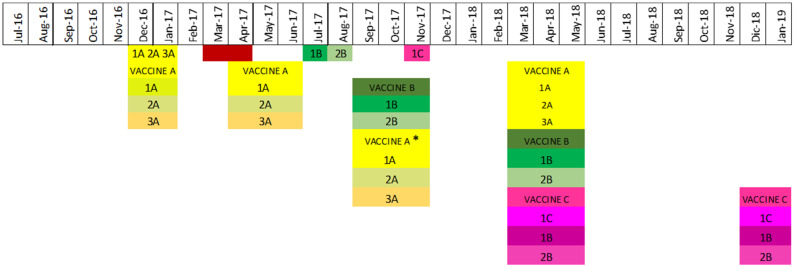
Vaccine use and influenza A virus strain inclusion throughout the study. Strains are denominated by the number/letter combination in each cell. Yellow squares represent circulating and selected strains included in vaccine A, green squares represent circulating and selected strains in vaccine B (administered in fall 2018), and pink squares represent vaccine C (administered in spring 2018). Red squares represent months when the circulating strains identified had an HA amino acid similarity less than 95% with the strains in vaccine A but were not selected as vaccine candidates. * Only two herds used vaccine A.

**Table 1 viruses-17-00240-t001:** Vaccine composition and season of administration for the three custom-made influenza A vaccines used in the study. * Number of herds using this vaccine.

Vaccine Name	Vaccine Code	Strain Code	Strain Collection Date	Subtype, US Clade, Global Clade (and Amino Acid Motif of H3)	Vaccine Administration/ Season
A	1317	1A	22-March-2016	H1N1, gamma, 1A.3.3.3	Fall 2016 (n = 30) * Spring 2017 (n = 32) Spring 2018 (n = 2)
		2A	20-January-2016	H1N2, delta2, 1B.2.1	
		3A	21-April-2016	H3N2, IVA, 3.1990.4.1 (NYKNYSS)	
B	1350 NEW	1B	10-January-2017	H1N2 delta2 1B.2.1	Fall 2017 (n = 31) Spring 2018 (n = 9)
		2B	17-November-2016	H3N2 IVA 3.1990.4.1 (KYNNYKY)	
C	1379	1C	02-May-2016	H1N1gamma 1A3.3.3	Spring 2018 (n = 21) Fall 2018 (n = 32)
		1B	10-January-2017	H1N2 delta 2 1B.2.1	
		2B	17-November-2016	H3N2 IVA 3.1990.4.1 (KYNNYKY)	

**Table 2 viruses-17-00240-t002:** Proportion of influenza A virus positive samples (or pools) three months before and after each vaccine administration. Letters A, B, and C represent the vaccines used. Results from first, second, and third vaccine administrations refer to testing of individual samples, while results from fourth and fifth vaccine administrations refer to testing of pooled samples. Results with *p* values < 0.05 are in bold font and indicate a statistically significant reduction in the proportion of IAV-positive samples after vaccination.

Vaccine Administration	Vaccination Date	Vaccine Used	No. Pos/No. Tot—Before (%)	No. Pos/No. Tot—After (%)	*p* Value
First	16-December	A	170/1050 (16.2)	125/965 (13)	0.381
Second	17-May	A	114/978 (11.7)	**26/975 (2.7)**	**0.003**
Third	17-October	B	30/975 (3.1)	67/1004 (6.7)	0.099
Fourth	18-March	A, B, C	30/975 (5.1)	**9/407 (2.2)**	**0.008**
Fifth	18-August	C	3/366 (0.8)	8/421 (1.9)	0.824

**Table 3 viruses-17-00240-t003:** Estimated proportions of influenza A-virus-positive samples in the farm system predicted by the generalized linear mixed model three months before and after administration of each of the vaccines. Each vaccine administration time includes the analysis for vaccinated and non-vaccinated (NO VACC) herds. Letters A, B, and C represent the codes of the vaccines used. Results with *p* values less than 0.05 are statistically significant. * NA: not available; model could not predict estimated proportions and 95% CI due to different sampling strategies before and after vaccination or due to farms without positive samples.

Vaccine Administration	Vaccination Year-Month	N Herds	Vaccine Used (Code)	Time Period Relative to Vaccination	Estimated Proportions of IAV- Positive Samples	95% CI	*p* Value
First	2016-December	6	NO VACC	Before	4.41%	0.80–21.11	<0.0001
				After	16.24%	3.32–52.31	
		29	A	Before	7.15%	2.96–16.27	<0.0001
				After	2.30%	0.89–5.78	
Second	2017-May	4	NO VACC	Before	1.78%	0.10–28.52	0.34
				After	0.70%	0.02–13.59	
		31	A	Before	1.99%	0.53–7.10	<0.0001
				After	0.30%	0.07–1.13	
Third	2017-October	2	NO VACC	Before	3.10%	0.05–63.66	0.99
				After	4.85%	0.09–72.55	
		31	B	Before	0.01%	0.02–0.75	<0.0001
				After	0.05%	0.08–2.75	
Fourth	2018-March	5	NO VACC	Before	NA *	NA	NA
				After	NA	NA	
		2	A	Before	2.68	0.03–68.90	0.80
				After	1.15	0.01–49.91	
		9	B	Before	NA	NA	NA
				After	NA	NA	
		18	C	Before	NA	NA	NA
				After	NA	NA	
Fifth	2018-August	1	NO VACC	Before	NA	NA	NA
				After	NA	NA	
		34	C	Before	NA	NA	
				After	NA	NA	

**Table 4 viruses-17-00240-t004:** Minimum and maximum nucleotide and amino acid percent identity of influenza A virus hemagglutinins comparisons of the strains detected in the swine farms and the strains contained in the vaccines. Each farm strain sequence was compared with the vaccine strain of the same H1 or H3 clade contained in the vaccine. NA *: not available, only one sequence to compare.

Vaccine Name	Vaccine Strain (Vaccine Letter Code)	Herd-Strain Subtype	% Nucleotide Identity		% Amino Acid Identity	
			Minimum	Maximum	Minimum	Maximum
A	H3N2 IVA (3A)	H3	94.35%	99.94%	95.05%	100.00%
B	H3N2 IVA (2B)	H3	86.28%	NA *	80.07%	93.98%
C	H1N1 gamma 1A3.3.3 (1C)	H1	88.54%	100.00%	90.64%	100.00%
A	H1N1 gamma 1A3.3.3(1A)	H1	90.24%	99.29%	91.70%	99.12%
C	H1N2 delta 2 1B.2.1 (1B)	H1	94.80%	99.86%	93.44%	100.00%
A	H1N2 delta 2 1B.2.1 (2A)	H1	94.35%	99.85%	93.98%	100.00%
B	H1N2 delta2 1B.2.1 (1B)	H1	96.09%	96.35%	95.93%	96.28%

**Table 5 viruses-17-00240-t005:** Minimum and maximum number of key amino acid (a.a.) differences and percent differences at the antigenic sites of the hemagglutinin protein of herd and vaccine strains of the same H1 or H3 clade. When only one ratio (and percentage) was reported, there was only one herd sequence of that clade to compare to the vaccine.

Vaccine Name	Vaccine Number Code	Vaccine Strain/Antigenic Site		Minimum Number of a.a. Differences/ Number of Key a.a. Sites Evaluated (%)	Maximum Number of a.a. Differences/ Number of Key a.a. Sites Evaluated (%)
A	1317	H3 IVA_3.1990.4.1		1/7 (14%)	5/7 (71%)
B	1350 NEW	H3 IVA_3.1990.4.1		4/7 (57%)	
C	1379	H1 gamma_1A.3.3.3	Sa antigenic site	2/13 (15%)	4/13 (31%)
			Sb antigenic site	2/13 (15%)	4/13 (31%)
			Cb antigenic site	2/6 (33%)	
			Ca1 antigenic site	2/11 (18%)	
			Ca2 antigenic site	0/9 (0%)	3/9 (33%)
A	1317	H1 gamma_1A.3.3.3	Sa antigenic site	2/13 (15%)	4/13 (31%)
			Sb antigenic site	2/13 (15%)	4/13 (31%)
			Cb antigenic site	1/6 (17%)	2/6 (33%)
			Ca1 antigenic site	0/11 (0%)	4/11 (36%)
			Ca2 antigenic site	0/9 (0%)	2/9 (22%)
A	1317	H1 delta2_1B.2.1		1/23 (4%)	13/23 (57%)
B	1350 NEW	H1 delta2_1B.2.1		1/23 (4%)	7/23 (30%)

**Table 6 viruses-17-00240-t006:** Summary of the key amino acid differences at each antigenic site by herd strain compared to each respective vaccine strain. The first letter in the amino acid differences column represents the circulating farm strain amino acid, the number represents the amino acid position (using mature protein numbering), and the second letter is the amino acid in the vaccine strain. * Number of sequences with the amino acid at that position/total number of sequences.

Strain/Clade		Amino Acid Differences	Number of Sequences/Total Sequences *
H3		N145K	8/10
		N156K	3/10
		K189S	5/10
		Y193S	3/10
H1 gamma_1A.3.3.3	Sa antigenic site	E155G	4/27
		Q163K	5/27
	Sb antigenic site	T185S	5/27
	Cb antigenic site	S71Y	2/27
	Ca1 antigenic site	D168N	5/27
		T203S	5/27
	Ca2 antigenic site	A141T	5/27
		K142N	12/27
H1 delta2_1B.2.1		G121D	10/27
		V152E	24/27
		A214V	13/27

## Data Availability

The 68 IAV HA nucleotide sequence data supporting the results and conclusions of this article will be made available by the authors on request.

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
