# Peer review of "Reduction of Influenza A Virus Prevalence in Pigs at Weaning After Using Custom-Made Influenza Vaccines in the Breeding Herds of an Integrated Swine Farm System"

_viruses, 2025, doi:10.3390/v17020240_

Round 1
Reviewer 1 Report
Comments and Suggestions for Authors
The manuscript titled "Reduction of influenza A virus prevalence in pigs at weaning after using custom-made influenza vaccines in the breeding herds of an integrated swine farm system" reported that how a comprehensive approach of virus surveillance and vaccination may reduce influenza A disease burden in weaning pigs. The study reported an interesting approach that may be useful for the stakeholders. The study is well designed and conducted. The manuscript is well written in all the aspects. This reviewer has no suggestions for the authors and congratulate them for this flaw-less investigation.
Author Response
Comment 1: The manuscript titled "Reduction of influenza A virus prevalence in pigs at weaning after using custom-made influenza vaccines in the breeding herds of an integrated swine farm system" reported that how a comprehensive approach of virus surveillance and vaccination may reduce influenza A disease burden in weaning pigs. The study reported an interesting approach that may be useful for the stakeholders. The study is well designed and conducted. The manuscript is well written in all the aspects. This reviewer has no suggestions for the authors and congratulate them for this flaw-less investigation.
Response 1: The authors think the reviewer for the kind comments. We appreciate the rapid review.
Reviewer 2 Report
Comments and Suggestions for Authors
This is a very structured and technically sound work.
Their measures and efforts are certainly to be made in this form if there are real clinical problems and production losses.
However, the use of commercial vaccines often appears to be more practical and pragmatic in everyday practice. Der Aufwand der Sie ausüben ist immens hoch. Das mögen Landwirte nur im absoluten Notfall mitmachen.
Also to note:
Graaf-Rau et al. Focus on other technical points that I also consider very important.
They describe that the age of the animals plays a role. They believe that it would be advisable to vaccinate younger animals as well.
In this way, they hope that rapid immunological stability and interruption of IAV transmissions will occur.
The colleagues work with experimentally produced and commercial vaccines. In this context, they shed light on one of the key points.
Intranasally administered vaccines cause a different immune response than intramuscularly administered vaccines and may therefore offer more effective protection, especially since in this case the vaccination route and infection route are the same.
Vaccines with and without adjuvant are used here. However, they note that the use of an adjuvant (as is present in the commercial vaccines) offers broader protection.
Perhaps the most practical way would be to use a more frequently revised vaccine commercial vaccine for everyday practice. Furthermore, the vaccination of different age groups should be considered.
Author Response
comment 1: This is a very structured and technically sound work.
response1: The authors thank the reviewer for the compliment.
comment 2: Their measures and efforts are certainly to be made in this form if there are real clinical problems and production losses.
response 2: we agree with the reviewer.
comment 3: However, the use of commercial vaccines often appears to be more practical and pragmatic in everyday practice.
response 3: We agree with the reviewer that vaccination use must be pragmatic and practical.
comment 4: Der Aufwand der Sie ausüben ist immens hoch. Das mögen Landwirte nur im absoluten Notfall mitmachen.
response 4: We respectfully disagree that farmers would only undertake an immense effort to vaccinate and perform surveillance for emergency reasons only. The farm system that was part of this study voluntarily performed vaccination and surveillance to control influenza and reduce prevalence and they continue to do so today. The farm system does influenza vaccination and surveillance routinely to increase pig health and to decrease influenza virus in their pigs, which in turn decreases opportunities for virus spread to other species.
comment 5: Also to note: Graaf-Rau et al. Focus on other technical points that I also consider very important.
response 5: the authors assume that this is the Graaf-Rau et al. reference. We thank the reviewer for mentioning it. Graaf-Rau A, Schmies K, Breithaupt A, Ciminski K, Zimmer G, Summerfield A, Sehl-Ewert J, Lillie-Jaschniski K, Helmer C, Bielenberg W, Grosse Beilage E, Schwemmle M, Beer M, Harder T. Reassortment incompetent live attenuated and replicon influenza vaccines provide improved protection against influenza in piglets. NPJ Vaccines. 2024 Jul 13;9(1):127. doi: 10.1038/s41541-024-00916-x. PMID: 39003272; PMCID: PMC11246437.
comment 6: They describe that the age of the animals plays a role. They believe that it would be advisable to vaccinate younger animals as well.
response 6: the authors agree that Graaf-Rau et al made this point about pig age.
comment 7: In this way, they hope that rapid immunological stability and interruption of IAV transmissions will occur.
response 7: the authors agree that Graaf-Rau et al had this as the goal of vaccination in their 2024 manuscript published in NPJ Vaccines.
comment 8: The colleagues work with experimentally produced and commercial vaccines. In this context, they shed light on one of the key points.
response 8: the authors agree that Graaf-Rau et al performed a good study in a controlled setting using killed, reassortment incompetent live attenuated, and replicon influenza vaccines. however, the authors of the viruses-3429868 manuscript under review here respectfully request to restrict the comments and responses to those regarding the use of killed autogenous vaccines to adult females on pig farms.
comment 9: Intranasally administered vaccines cause a different immune response than intramuscularly administered vaccines and may therefore offer more effective protection, especially since in this case the vaccination route and infection route are the same.
response 9: the authors did not have access to intranasal vaccines and did not evaluate them.
comment 10: Vaccines with and without adjuvant are used here. However, they note that the use of an adjuvant (as is present in the commercial vaccines) offers broader protection.
response 10: the authors respectfully disagree with this comment. All vaccines used in the Garrido-Mantilla et al manuscript being reviewed here as the viruses-3429868 manuscript were adjuvanted. see lines 269-271 for details on the adjuvant.
comment 11: Perhaps the most practical way would be to use a more frequently revised vaccine commercial vaccine for everyday practice.
response 11: the authors agree that using a more frequently revised vaccine commercial vaccine would be more practical. unfortunately, in the USA, revising the autogenous vaccines every 6 months is the only feasible option due to the fact that commercial vaccine licensing requirements take at least two years to fulfill. See in the authors submitted manuscript our reference #7 or read the regulation here https://www.aphis.usda.gov/sites/default/files/memo_800_111.pdf.
comment 12: Furthermore, the vaccination of different age groups should be considered.
response 12: the authors agree that future vaccination programs for this large farm system might consider vaccinating different age groups. However, given the size of the system (35 sow farms with an average of 3000 sows per farm), the number of pigs to be vaccinated would be at a minimum 1,050,000 pigs (=35 farms X 3000sows per farm X10 pigs per sow) and such a number would likely be considered cost-prohibitive. Therefore, we studied, researched, and submitted this manuscript which focused on adult sow vaccinations.
Reviewer 3 Report
Comments and Suggestions for Authors
This manuscript describes on efficacy for the custom made influenza vaccines in swine farms. Here are some comments.
1. Fig 1. Better describes with graph to easy understand. It is hard to understand in this form.
2. Table 1. What is the difference between vaccines name A and C? Their subtype, clade seems same but only difference were name, code and strains name. Better clarify.
3. Table 2. Vaccines were administrated as the combination like A, A, B and A,B,C. Can we say those reduction of positive by vaccination was by vaccination A or B or C? Hard to understand study design.
4. Table 3. NA for those fourth and fifth vaccination in the result table. What is the meaning on this?
Author Response
comment 1: Fig 1. Better describes with graph to easy understand. It is hard to understand in this form.
response 1: the authors thank the reviewer for the suggestion. To improve the figure, we have made it a color figure and the color figure has been attached for your review. the legend for figure 1 has been revised. Specifically, figure 1 legend revised lines are 303-313 are, "Figure 1. Influenza A virus (IAV) farm status assessed by IAV rRT-PCR test results per farm (rows) by month (columns) from July 2016 to January 2019. A farm was considered to be of positive status during that month if at least one sampling event was positive. Red cells indicate positive results for the farm and the farm is given an influenza positive status for that month. Green cells indicate negative results for the farm that month and the farm is given influenza negative status; white cells indicate no samples were collected from that farm on that month. Months (columns) in bold italics with cyan highlighting indicate vaccination administration times. Herd prevalence was calculated by dividing the number of positive herds (if at least one sample tested positive) by the total number of sampled herds in a month and is given at the bottom of each column for each month."
comment 2Table 1. What is the difference between vaccines name A and C? Their subtype, clade seems same but only difference were name, code and strains name. Better clarify.
response 2: we have revised table 1 to include the strain collection date. In addition, supplementary Figures 1 and 2 show the phylogenetic differences of the strains (vaccines are in purple on the trees). We hope this clarifies that the strains were indeed different.
comment 3: Table 2. Vaccines were administrated as the combination like A, A, B and A,B,C. Can we say those reduction of positive by vaccination was by vaccination A or B or C? Hard to understand study design.
response 3: the authors apologize for the confusion. it may be easier to understand the study if one focuses on the date of vaccine administration and the vaccine used at that date. also, viewing figure 2 alongside table 2 is recommended in order to visualize the study chronologically (figure 2) and the influenza testing results by administration date. The vaccination program we evaluated in this study had three customized vaccines (A, B and C) each with a different IAV strain composition. Vaccines were administered at 5 different time points (dates) - first, second, third, fourth, and fifth. The vaccine selected for administration at those time points (dates) was based on the infection challenges identified via the surveillance program. More specifically, at the first administration time point in December 2016, all farms used Vaccine A; at the second administration time point in May 2017, all farms again used Vaccine A; at the third administration time point in October 2017, 33 farms used Vaccine B and 2 used Vaccine A; at the fourth administration time point in March 2018, farms used either Vaccine A, Vaccine B, or Vaccine C depending on their surveillance results (for example, if surveillance results showed that the farm was having infections with H3N2 IVA with asparagine (N) at key amino acid position 145 and also H1N1 gamma infections, that farm would use Vaccine A. If a farm was having H1N2 delta 2 infections and having infections with H3N2 IVA with lysine (K) at key amino acid position 145, that farm would use vaccine B There was a reduction in influenza after the first, second, and fourth administration of vaccines.
comment 4: Table 3. NA for those fourth and fifth vaccination in the result table. What is the meaning on this?
response 4: NA means that predicted estimated proportions and 95% CI are Not Available. In the legend for table 3, lines 437-439, we defined NA as follows. "NA: not available; model could not predict estimated proportions and 95% CI due to different sampling strategies before and after vaccination or due to farms without positive samples."
